# Implementation of the Homogenization Method in the Numerical Estimation of Wafer Warpage

**Soufyane Belhenini** [1,*] **, Imad El Fatmi** [1] **, Caroline Richard** [2] **, Abdellah Tougui** [3] **and Fabrice Roqueta** [4]

1 Smart Structures Laboratory, University of Ain Temouchent, BP 284 Route S.B.A, Ain Temouchent 46000, Algeria
2 GREMAN UMR CNRS 7347, Université de Tours, Insa Centre Val de Loire, 37200 Tours, France
3 Laboratory of Mechanics Gabriel Lamé, University of Tours, 37200 Tours, France
4 STMicroelectronics, 10 Rue Thalès de Milet CS 97155, 37071 Tours, France
* Correspondence: soufyane.belhenini@univ-temouchent.edu.dz

**Abstract:** Given the growing global demand for high-performance microcomponents, while keeping the size of the microcomponents as small as possible, several manufacturers have chosen to increase the number of thin layers to increase the integration density. These thinner layers cause warping-type deformations during processing. In this study, warping during the development of a stacking composed of a silicon substrate coated with two thin layers, one dielectric in undoped silicate glass (USG) and the other metallic in platinum, was numerically analyzed and validated by comparison with experimental measurements. The numerical study presented in this paper has several components that make it simple and reliable. Indeed, simplifications of the loading conditions were introduced and validated by comparison with experimental results. Another part of the simplification is to integrate a homogenization approach to reduce the number of calculations. The efficiency and precision of the homogenization approach were validated for the mechanical and thermomechanical models by comparing the heterogeneous and homogenized models.

**Keywords:** wafer warpage; finite element modeling; homogenization; residual stress

## 1. Introduction

Electronic microcomponents are manufactured from hundreds or even thousands of wafers that are usually made of silicon. The manufacturing process involves the deposition of thin metal and dielectric films on silicon substrates. The difference in the mechanical properties, particularly the coefficients of thermal expansion between the materials involved in the stack, causes cracks and even delamination. The deflections of the pile-up induce mechanical fatigue during the lifetime and represent a limiting factor for reliability. Deflection control is necessary before large-scale production can begin. Several techniques have been developed to measure stack warping. Spectroscopic ellipsometry is a non-destructive technique used to measure the curvature of a sample by measuring the polarization of a beam at a reflection on a surface [1]. The X-ray diffraction technique allows the deformation of the measured network to be measured. This technique is only valid for crystalline materials [2]. Conventional 3D DIC (3D digital image correlation), a non-contact full-field optical technique, is also commonly used for measuring wafer deflection. Yao et al. [3] measured the curvature of a Si/USG/Pt stack using capacitive gages. This instrument consists of 33 capacitive sensor pairs integrated into radial geometry. Each pair of sensors measures the local displacement of the stack by measuring the distance between the median surface of the plate and the reference plane, as shown in Figure 1. Wafer warping is defined as total warping, which is the sum of the absolute values of the maximum and minimum local warping. The experimental technique usually requires preliminary preparation of the samples. This can take a long time, which slows down

the development projects for new microcomponents. Researchers have turned to other techniques for estimating warping, such as semi-empirical [4,5] and numerical approaches.

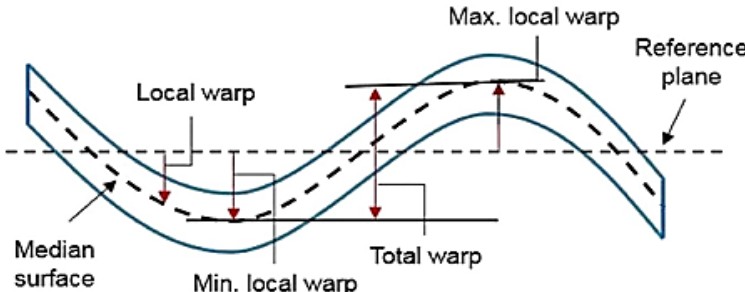

**Figure 1.** Illustration of the warping measurement method [3].

Recently, progress in the electronics industry has made electronic products more miniature and efficient, which has increased the number of thin layers that form stacks, making them susceptible to deformation. Deformation of the wafer increases because of the heterogeneous nature of the thin films and the difference in the thermal expansion coefficient. The stress induced by wafer curvature is one of the root causes of process and device failure, such as delamination, cracking, and decreased performance [6].

Numerical modeling can be used to evaluate the curvature. This approach encounters difficulties when used to solve multiscale problems owing to the huge need for computer memory and CPU time. However, in engineering, it is sufficient to obtain macroscopic solutions for multi-scale material structures. Therefore, determining the macroscopic effective properties of heterogeneous materials has become an important problem in many engineering applications [7].

Recently, several multiscale methods have been developed as part of the small-deformation elasticity or elastoplastic theory for heterogeneous materials [8]. The macroscopic effective constitutive response is predicted as a result of the analytical or numerical solution of a boundary-value problem at the microscopic level. For analytical methods, for example, the Eshelby method [9] considers the form of inhomogeneity using the Eshelby tensor and proposes an equivalent inclusion method. This approach was developed by several authors such as Mori and Tanaka [10], Hashin and Strikman [11], and Hill [12]. These methods allow for prediction of the equivalent properties of constituent materials for fairly simple geometries and low-density fractions. However, their limits do not stop here; they cannot describe the evolution of stresses and macroscopic deformations for complex structures. To overcome the difficulties of numerical homogenization approaches, the representative volume element (RVE) method was developed [13,14]. Babuska [15] developed an asymptotic computational homogenization method. Miled et al. [16] developed an analytical multi-step homogenization method to estimate the deflection of a silicon wafer coated with three thin films. Cheng et al. [17] used the homogenization method with a submodelling technique to estimate the warpage of a stack of five layers with silicon vi (TSV).

Yao et al. [3] proposed a numerical model for determining the warping of two thin layers stacked on a silicon substrate. The stack consisted of an undoped silicate glass USG insulating layer and a platinum Pt metallic layer. The model developed by Yao et al. [3] consists of simulating the entire multi-step thermal cycle and activating and deactivating layers.

The current study constitutes a contribution in the simplification of numerical estimation of wafer warping. The numerical approach consists of simplifications of the numerical approach developed by Yao et al. [3]. The homogenization method was used to replace the real structure with a silicone substrate and two thin films (USG and Pt) into a simplified structure with a substrate and an equivalent thin film. The equivalent properties are calculated by applying the mixture laws that are widely used to simulate composite materials. A simplified loading condition for the numerical simulation was also suggested. For the loading condition, the proposed assumption used the final imposed thermal cycle step

instead of the complete cycle. Note that the final warping was largely the result of the final cooling step. This assumption was validated through comparing the obtained results with those obtained by Yao et al. [3]. This contribution constitutes support for future research projects for the numerical estimation of silicon wafers with very complex geometries, in particular wafers used for 3D microcomponents.

## 2. Materials and Methods

For the first simplification verification, a 2D axisymmetric model was created using the Abaqus code for the estimation of the Si/USG/Pt stack warpage. The geometrical model and boundary conditions are shown in Figure 2. Three Pt thicknesses were used (100, 150, and 300 nm). A roller boundary condition was applied to the revolution axis. A tie contact was used for all of the contact areas. The material properties are listed in Table 1.

Two thermal cycles were used as loading conditions. The first, the whole cycle, is the one employed by Yao et al. [3] with successive activation of the layers, as shown in Figure 3. The second consists of the use of the latest step of the cycle imposed by Yao et al. [3] (cooling from 450 °C to 22 °C). The warpage obtained using the two models was compared to validate the first simplification.

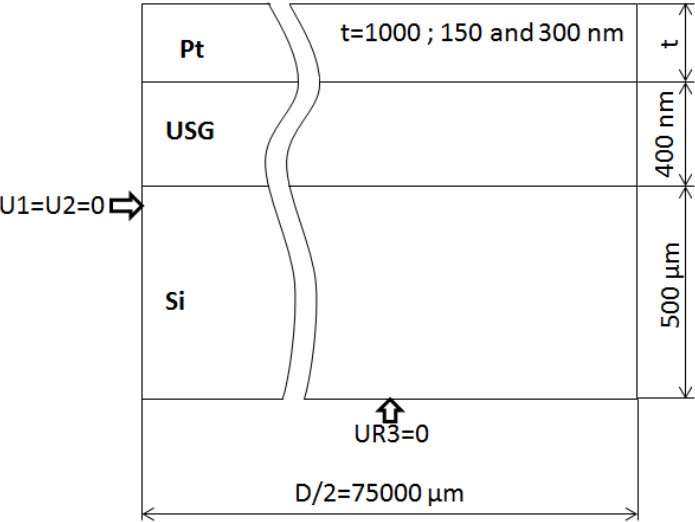

**Figure 2.** Geometry and boundary conditions of the 2D axisymmetric model.

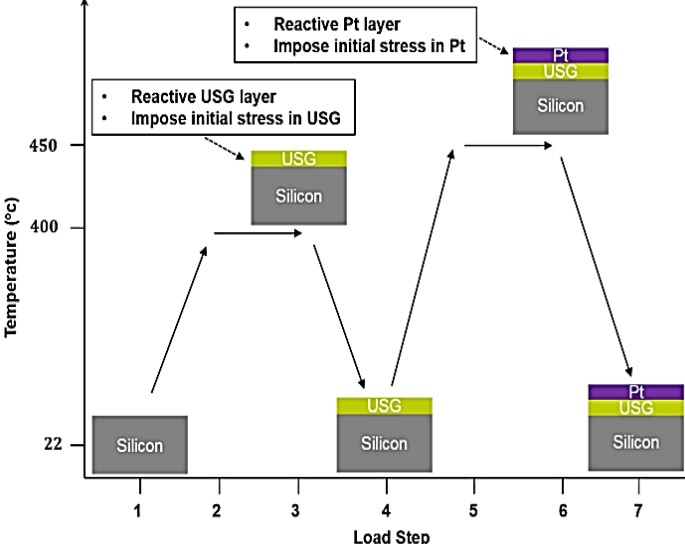

**Figure 3.** The whole thermal cycle used by Yao et al. [3].

**Table 1.** Materials' properties.

|  | USG | Pt | Si |
|---|---|---|---|
| E (GPa) | 94 | 177 | 169 |
| $\upsilon$ | 0.17 | 0.39 | 0.063 |
| $\alpha$ (10$^{-6}$/°C) | 0.5 | 9 | 2.8 |

For the second simplification, using the homogenization approach, the wafer warpage obtained by the full heterogeneous model was compared with that obtained by the homogeneous model. Figure 4 shows the heterogeneous and equivalent homogenized models. The equivalent material is assumed to be transversally isotropic.

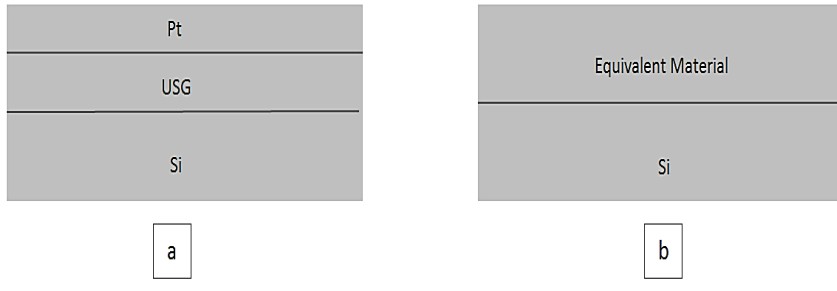

**Figure 4.** (**a**) Heterogeneous model and (**b**) equivalent homogenized model.

The mechanical properties of the equivalent model were calculated using the following equations derived from mixture laws [18]:

Longitudinal Young's modulus $E_{1,2}$, Equation (1):

$$E_{1,2} = (E_{Pt} \cdot V_{Pt}) + (E_{USG} \cdot V_{USG}) \tag{1}$$

With:

$E_{1,2}$: Equivalent longitudinal Young's modulus
$E_{Pt}$: Young's modulus of platinum
$E_{USG}$: Young's modulus of USG
$V_{Pt}$: volume fraction of Pt
$V_{USG}$: volume fraction of USG
Transverse Young's module $E_3$, Equation (2):

$$\frac{1}{E_3} = \frac{V_{Pt}}{E_{Pt}} + \frac{V_{USG}}{E_{USG}} \tag{2}$$

With:

$E_3$: Equivalent transverse Young's modulus.
$E_{Pt}$: Young's modulus of platinum.
$E_{USG}$: Young's modulus of USG.
$V_{Pt}$: volume fraction of Pt.
$V_{USG}$: volume fraction of USG.
The equivalent Poisson coefficients are as follows:
The Poisson's ratio $v_{12}$, Equation (3):

$$v_{12} = (V_{Pt} \cdot v_{Pt}) + (V_{USG} \cdot v_{USG}) \tag{3}$$

And Equation (4)

$$\frac{E_3}{v_{13}} = \frac{E_1}{v_{12}} \tag{4}$$

With:

$V_{Pt}$: volume fraction of Pt.

$V_{USG}$: volume fraction of USG.
$\nu_{Pt}, \nu_{USG}$ : Poison coefficients of Pt and USG.
Shear modulus Equation (5):

$$G_{12} = \frac{E_1}{2(1 + \nu_{12})} \tag{5}$$

With:
$E_1$: Equivalent Young's modulus.
$\nu_{12}$: Poisson's ratio.
And Equation (6):

$$G_{13} = G_{23} = \frac{E_3}{2(1 + \nu_{13})} \tag{6}$$

With:
$E_3$: Equivalent transverse Young's modulus.
$\nu_{13}$: Poisson's ratio.
We used the expression proposed by Shapery et al. [19] to calculate the coefficient of longitudinal thermal expansion Equation (7):

$$\alpha_{1,2} = \frac{E_{Pt}\alpha_{Pt}V_{Pt} + E_{USG}\alpha_{USG}V_{USG}}{E_{Pt}V_{Pt} + E_{USG}V_{USG}} \tag{7}$$

With:
$\alpha_1$: coefficient of equivalent longitudinal thermal expansion.
$E_{Pt}, E_{USG}$: Young's moduli of Pt and USG, respectively.
$\alpha_{Pt}, \alpha_{USG}$: coefficient of thermal expansion of Pt and USG, respectively.
$V_{Pt}, V_{USG}$: volume fraction of Pt and USG, respectively.
The transverse thermal expansion is estimated as Equation (8):

$$\alpha_3 = (\alpha_{USG}V_{USG} + \alpha_{Pt}V_{Pt}) + \frac{(E_{USG}\nu_{Pt} - E_{Pt}\nu_{USG})}{E_{Pt}V_{Pt} + E_{USG}V_{USG}}V_{USG}V_{Pt}(\alpha_{USG} - \alpha_{Pt}) \tag{8}$$

The properties of the homogenized material are listed in Table 2.

**Table 2.** Equivalent material properties.

|  | Pt (100 nm) | Pt (150 nm) | Pt (300 nm) |
|---|---|---|---|
| $E_1, E_2$ (GPa) | 110.600 | 116.660 | 129.430 |
| $E_3$ (GPa) | 103.740 | 107.800 | 117.924 |
| $\nu_{12}, \nu_{13}$ | 0.214 | 0.23 | 0.26 |
| $\nu_{23}$ | 0.2 | 0.212 | 0.23 |
| $G_{12}, G_{13}$ (GPa) | 45.551 | 47.430 | 51.360 |
| $G_{23}$ (GPa) | 43.225 | 44.472 | 47.930 |
| $\alpha_1, \alpha_2$ ($10^{-6}/\,°C$) | 3.22 | 4.02 | 5.47 |
| $\alpha_3$ ($10^{-6}/\,°C$) | 4.31 | 4.96 | 6.08 |

## 3. Results and Discussions

### 3.1. Validation of the Loading Simplification

To examine the accuracy of the numerical results obtained using the last step of the thermal cycle, we compared them with the results obtained using the complete thermal cycle and experimental measurements of the deflection. Figure 5 shows the wafer warpage obtained by applying a complete thermal cycle for three Pt films thicknesses (100, 150, and 300 nm). The U2 displacement values are taken at a distance of 68 mm from the center of the wafer in order to be able to compare the results obtained numerically with those measured experimentally. Note that the curvature measuring machine takes measurements at this location [3]. Figure 5 shows an increase in the deflection as the thickness increases. The

results obtained by applying the full thermal cycle have a negligible difference compared with the results obtained by the last stage of the complete thermal cycle shown in Figure 6. The measured deflection values were compared to those obtained numerically. Figure 7 shows the high degree of agreement between the two numerical approaches. It should be noted that an average difference of 19% existed between the numerical and experimental results. This difference is mainly caused by ignoring the intrinsic stresses induced in the thin layers during processing. Therefore, it can be concluded that the simplification of the complete thermal cycle of loading into a simplified cycle involving only the last cooling phase has virtually no impact on the numerical results.

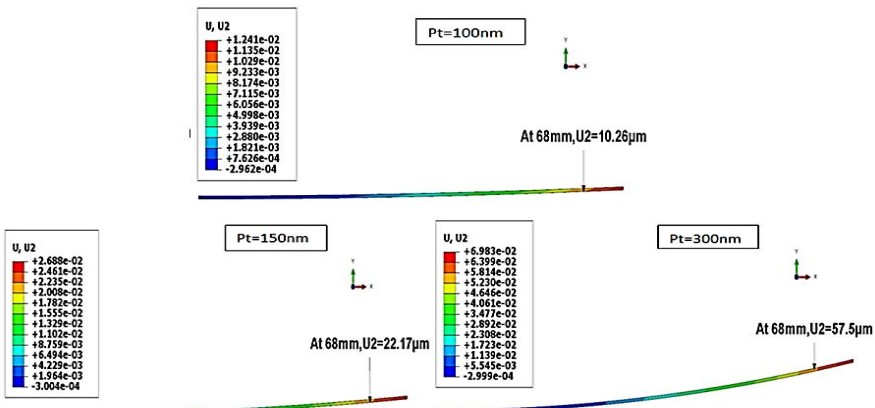

**Figure 5.** Warpage of the Si/USG/Pt stack using the full thermal cycle load.

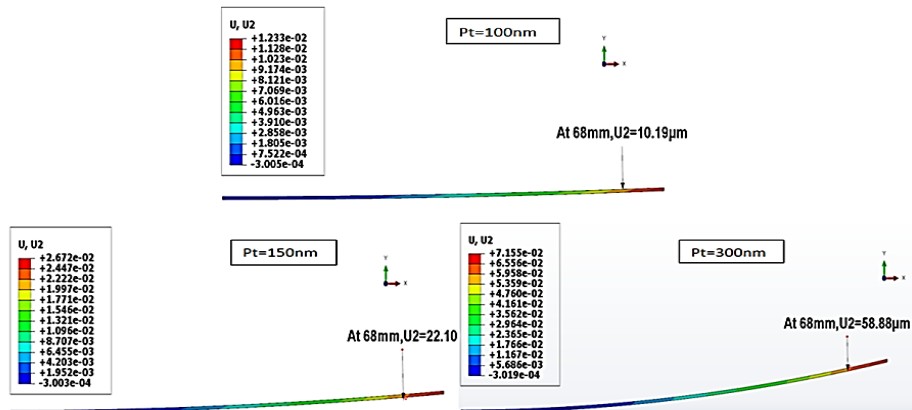

**Figure 6.** Warpage of the Si/USG/Pt stack using the last step of the full thermal cycle.

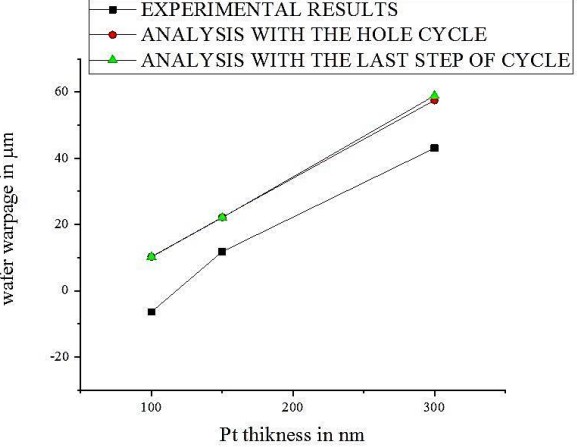

**Figure 7.** Numerical results and experimental measurements [3] of Si/USG/Pt warpage.

### 3.2. Validation of the Homogenization Approach

We have developed an axisymmetric 2D model that consists of a 500 μm Si substrate on which a homogeneous film is deposited, which represents the two thin films (USG + Pt) with the properties listed in Table 2. The imposed thermal load represents the last stage of the complete thermal cycle (cooling from 450 to 22 °C). The results obtained using the homogenized model are shown in Figure 8. The evolution of the defect as a function of the Pt layer thickness obtained by the homogenized model was compared with that obtained by the heterogenized model (Figure 9). It can be observed that the two curves have similar shapes. A 5% discrepancy was found between the warpage obtained by the homogenized model and that of the heterogeneous model for the 100 nm Pt layer. For the 150 nm Pt layer, the difference between the two models is approximately 11.53%, which increases to 19.63% for the models with a Pt layer of 300 nm. It can be seen that the difference between the warpage obtained by the two models increases with increasing Pt thickness. The mean difference between the homogenized model results and the heterogeneous model was 12.05%. It can be concluded that the homogenized approach presented in this study is reliable for small layer thickness. For thick layers, it was necessary to develop a formula for the transverse thermal expansion coefficient.

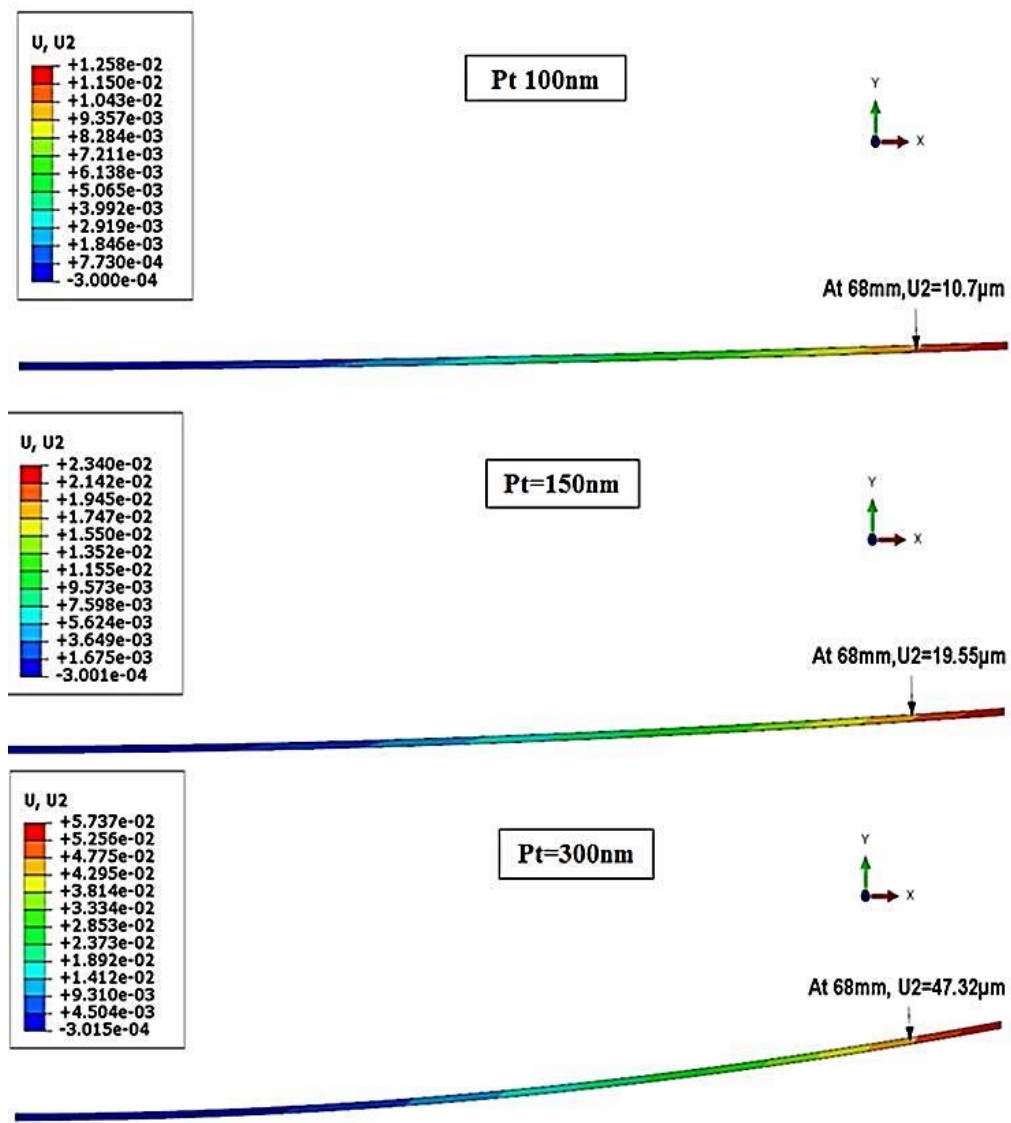

**Figure 8.** Warpage of the homogenized stack.

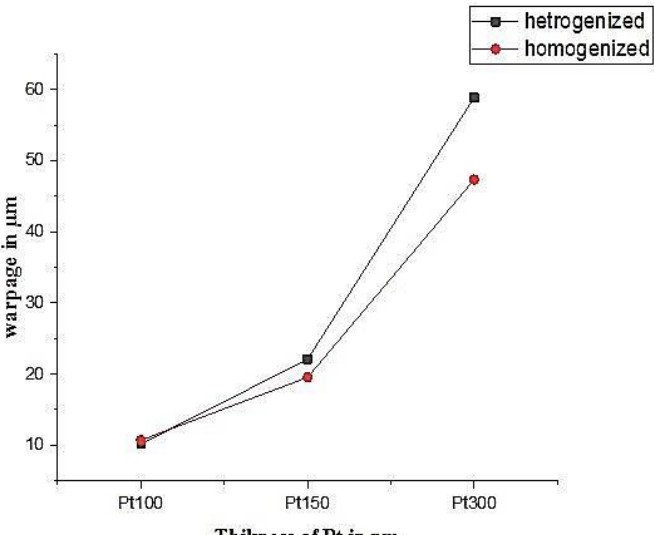

**Figure 9.** Warpage obtained using the homogenized/heterogeneous model.

### 4. Conclusions

In this study, a method for simplifying the numerical estimation of warping at the level of silicon platelets coated with two thin layers was developed and validated. The first simplification replaces the complete thermal cycle with the last phase of the thermal cycle in the loading case. The second simplification is the use of the homogenization method to replace the two thin films with a single film with equivalent properties.

The obtained results can be summarized as follows:

- The replacement of the complete thermal cycle with the last cooling phase had virtually no effect on the numerical estimates of the warps.
- The evolution of warping as a function of the thickness of Pt obtained numerically presents the same appearance as that measured experimentally.
- Increasing the thickness of the metal film increased the warping.
- The difference between the numerical and experimental results is mainly owing to the non-inclusion of intrinsic stresses in the numerical approaches.
- The homogenization technique made it possible to reliably estimate the wafer warpage.

**Author Contributions:** Conceptualization, S.B.; methodology, S.B.; software, I.E.F.; validation, S.B.; formal analysis, S.B. and I.E.F.; investigation, I.E.F.; writing—original draft preparation, I.E.F. and S.B.; writing—review and editing, C.R. and F.R.; visualization, C.R.; supervision, S.B., A.T. and F.R. All authors have read and agreed to the published version of the manuscript.

**Funding:** This research received no external funding.

**Institutional Review Board Statement:** Not applicable.

**Informed Consent Statement:** Not applicable.

**Data Availability Statement:** Not applicable.

**Acknowledgments:** The authors gratefully acknowledge the assistance of the director and technical staff of the Smart Structures Laboratory at Ain Temouchent University. Yao Wei Zhen is most grateful for his support. The authors are also grateful to all those who contributed to the realization of this work, particularly the CAD staff at STMicroelectronics Tours and GREMAN Tours researchers.

**Conflicts of Interest:** The authors declare no conflict of interest.

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
