# Peer review of "Implementation of the Homogenization Method in the Numerical Estimation of Wafer Warpage"

_coatings, doi:10.3390/coatings13020318_

Round 1

Reviewer 1 Report

The paper investigates the warping with the development of a stacking composed of a silicon substrate. The paper needs some revision before being accepted for publication through the following points 

1- I think the title of the paper needs to be changed to a more suitable title clarifying the content of the paper. 

2- There are some grammatical errors that need to be revised. Please revise the paper for grammar and punctuation. 

3- at the end of page 1, please write the citation of ref [4] and [5] to [4, 5]. 

4- The resolution of the figures needs to be enhanced to a better resolution. 

5- The novelty of the paper needs to be clearly stated in the introduction part in a separate paragraph. 

6- The organization of the paper is missing please add the organization part. 

7- Eq. (1) to (3) where it has been derived form and ifthery exist please add some referamce 

8- Figure 6 need to be regraphed since it can not be seen in this form. 

9- In the simulator section, what method is used to validate the numerical results the author did not mention it ? 

10- the funding statement and the acknowledgment section need to be rewritten since it contains some wrong words. 

11- The reference list is out of date since no recent references are available. The authors need to add some recent references. 

after addressing these  major points the paper can be reconsidered for  publication 

Author Response

The authors would like to thank the reviewer for the quality of the report and the very pertinent remarks highlighted. The collaboration of the reviewer will certainly contribute to delivering a high quality document.

1- I think the title of the paper needs to be changed to a more suitable title clarifying the content of the paper. 

The title has been changed to “Implementation of the homogenization method in the numerical estimation of wafer warpage”.

2- There are some grammatical errors that need to be revised. Please revise the paper for grammar and punctuation. 

Thanks for the comment, the errors have been revised.

3- at the end of page 1, please write the citation of ref [4] and [5] to [4, 5]. 

Thanks for the comment, the writing of the citation has been changed.

4- The resolution of the figures needs to be enhanced to a better resolution. 

The figures' resolution has been improved

5- The novelty of the paper needs to be clearly stated in the introduction part in a separate paragraph. 

A separate paragraph outlining the originality of the study has been included in the introduction.

6- The organization of the paper is missing please add the organization part.

 The journal does not require a separate section for the organization of the article. The organization has been presented in the abstract and in the introduction.

7- Eq. (1) to (3) where it has been derived form and if thery exist please add some referamce 

A reference has been included.

8- Figure 6 need to be regraphed since it can not be seen in this form.

 The figure has been improved

9- In the simulator section, what method is used to validate the numerical results the author did not mention it ? 

The validation of the numerical approach was carried out by comparing the obtained numerical results with experimental measurements. This comparison is presented in Figure 7.

10- the funding statement and the acknowledgment section need to be rewritten since it contains some wrong words.

 Thanks for the comment, the correction has been made.

11- The reference list is out of date since no recent references are available. The authors need to add some recent references. 

Recent references have been added.

Reviewer 2 Report

The authors proposed two simplifications including thermal cycle loading and the homogenization method for the multi-layered thin film structure. However, some important problems should be addressed.

1. Why did the authors apply the simplification for the multi-layered structure instead of evaluating the mechanical performance with traditional FEM method as the calculation cost was not high for the thin structure with less elements?

2. In Fig. 7, the experimental results had large deviation with the analysis. The authors should improve the simplification for the thin structure as the simplification did not predict the mechanical behavior well.

3. In Fig. 9, the estimation differences between the heterogenized and homogenized structures were very large, for example, larger than 33% for Pt300. The authors should explain the large evaluation error as such large error is unacceptable.

4. There are many writing errors in the manuscript. The authors should carefully revise the sentences.

Author Response

The authors would like to thank the reviewer for the quality of the report and the very pertinent remarks highlighted. The collaboration of the reviewer will certainly contribute to delivering a high quality document.

  1. Why did the authors apply the simplification for the multi-layered structure instead of evaluating the mechanical performance with traditional FEM method as the calculation cost was not high for the thin structure with less elements?

Indeed, the simulation of the complete model does not present a relatively high calculation cost. However, our aim was to demonstrate that the mixture laws initially developed for composite materials can be used for modelling microelectronic microstructures. We specify that our study can be used as a starting point to model silicon wafers with much more complex structures, in particular that used to elaborate the 3D microcomponents (vertical connections TSV, several thin layers, geometric discontinuity in layers...).

  1. In Fig. 7, the experimental results had large deviation with the analysis. The authors should improve the simplification for the thin structure as the simplification did not predict the mechanical behavior well.

The difference between the numerical and the experimental results is principally due to not taking into account the intrinsic stresses caused by the manufacturing process. Several works use a compressive stress initial condition to approximate the numerical conditions to the experimental measurements.

Miled, H. F., Roqueta, F., Craveur, J. C., Le Bourhis, E., Gardes, P. and Tougui, A. Analytical multi-step homogenization methodology for a stack of thin films in microelectronics. 21st International Conference on Thermal, Mechanical and Multi-Physics Simulation and Experiments in Microelectronics and Microsystems (EuroSimE). Cracow, Poland, 2020, pp. 1-7, doi: 10.1109/EuroSimE48426.2020.9152652.  

Yao, W. Z., Roqueta, F., Craveur, J. C., Belhenini, S., Gardes, S.P. and Tougui, A. Modelling and analysis of the stress distribution in a multi-thin film system Pt/USG/Si », Mater. Res. Express. 2018, 5, no 4, 046405. doi: 10.1088/2053-1591/aaba4b.

We have opted to not force the model to reproduce exactly the experimental results, especially since our approach consists of comparing the numerical with the numerical (complete model and its homogenised equivalent).

  1. In Fig. 9, the estimation differences between the heterogenized and homogenized structures were very large, for example, larger than 33% for Pt300. The authors should explain the large evaluation error as such large error is unacceptable.

Thanks for the comment but the difference is less than 20%.  The warping for the heterogenous model for a Pt of 300 nm is estimated to be 58.88µm (please refer to figure 6). While the homogenized model presents a warpage of 47.32 µm (please refer to figure 8).

The difference is estimated by : u=100-[(47.32*100)/58.88]=19.63%

The mean difference between the homogenized model results and the heterogeneous model was 12.05% which constitutes an acceptable error.

  1. There are many writing errors in the manuscript. The authors should carefully revise the sentences.

Corrections have been implemented.

Reviewer 3 Report

A good piece of work. Numerical estimation of warping at the level of silicon platelets coated with two thin layers has been presented in a relatively simple way. The results of two homogenous and heterogeneous models were compared with some experimental values. The paper has been written well. English level is also acceptable. I have two other comments:

1. Table 1: Mention the description of each property in the footnote. 

2. the authors need to mention why more experimental results were not used for their model validation.

Based on the above, I recommend its publication in Coatings. 

Round 2

Reviewer 1 Report

The author has addressed all the points raised in the review report. therefore the paper can now be accepted for publication. 

Reviewer 2 Report

The paper can be published in current version.